# Elucidating the role of metal ions in carbonic anhydrase catalysis

Jin Kyun Kim [1], Cheol Lee [1], Seon Woo Lim[1], Aniruddha Adhikari[1], Jacob T. Andring[2], Robert McKenna[2], Cheol-Min Ghim [1] & Chae Un Kim [1]✉

Why metalloenzymes often show dramatic changes in their catalytic activity when subjected to chemically similar but non-native metal substitutions is a long-standing puzzle. Here, we report on the catalytic roles of metal ions in a model metalloenzyme system, human carbonic anhydrase II (CA II). Through a comparative study on the intermediate states of the zinc-bound native CA II and non-native metal-substituted CA IIs, we demonstrate that the characteristic metal ion coordination geometries (tetrahedral for $Zn^{2+}$, tetrahedral to octahedral conversion for $Co^{2+}$, octahedral for $Ni^{2+}$, and trigonal bipyramidal for $Cu^{2+}$) directly modulate the catalytic efficacy. In addition, we reveal that the metal ions have a long-range (~10 Å) electrostatic effect on restructuring water network in the active site. Our study provides evidence that the metal ions in metalloenzymes have a crucial impact on the catalytic mechanism beyond their primary chemical properties.

[1] Department of Physics, Ulsan National Institute of Science and Technology (UNIST), Ulsan 44919, Republic of Korea. [2] Department of Biochemistry and Molecular Biology, College of Medicine, University of Florida, Gainesville, FL 32610, USA. ✉email: cukim@unist.ac.kr

Metalloproteins are ubiquitous in nature and play indispensable roles in key biological processes, such as DNA synthesis, chemical signaling, and cellular metabolism[1,2]. Due to their versatile chemical reactivity (acidity, electrophilicity, and/or nucleophilicity), incorporated metal ions add functionality to proteins and help catalyze some of the most intricate reactions in nature[3,4]. The issues of metal binding affinity and specificity of metal ions to proteins have been studied based on the metal coordination stereochemistry[5,6] and semi-empirical and qualitative theories such as hard and soft acids and bases principle of Parr and Pearson[7] and Irving–Williams series of divalent ion stability[8,9]. However, the role of metal ions in the functioning of proteins and the metal–protein relationships remain unclear at the atomic level. For example, metalloenzymes substituted by non-native metal ions often exhibit drastically different catalytic activities[10,11], even when the substituted metal ions show chemical features broadly similar to the native one, such as ionic charge/size/mass, redox potential, electronic configuration, and allowed coordination geometry.

Among the various types of metalloenzymes, carbonic anhydrase (CA), the first enzyme recognized to contain zinc, is ubiquitous across all kingdoms of life and one of the most catalytically efficient enzymes ever known[12–16]. The enzyme catalyzes the reversible hydration of carbon dioxide ($CO_2$) and thereby plays a critical role in respiration, particularly in the $CO_2$ transport by way of blood-dissolved bicarbonate ($HCO_3^-$), and in intracellular pH homeostasis by maintaining $CO_2/HCO_3^-$ equilibrium. Within the wide classes of CA, CA II from human is well-suited to serve as a model system for investigating the role of metal ions because its overall structure is well-refined with atomic resolution (~1.0 Å). It possesses a well-defined active site containing a single metal-binding site (Fig. 1a, b), and the kinetic rates and fine details of the enzymatic mechanism have been studied extensively[17–21] (Fig. 1c).

The active site of CA II is located at the base cavity of a 15 Å depth from the surface and is further subdivided into three regions comprised of hydrophobic and hydrophilic regions, with an entrance conduit (EC) in-between[22–25] (Fig. 1a). These regions are responsible for substrate binding, proton transfer, and substrate/product/water exchange during catalysis, respectively (Fig. 1b). The active site zinc ion is tetrahedrally coordinated to the protein by the imidazole groups of three histidine residues, with the remaining tetrahedral site occupied by a solvent molecule (water or hydroxide ion, depending upon pH). The catalytic zinc ion in CA II serves as a Lewis acid; its primary role is to lower the $pK_a$ of the Zn-bound water from 10 to 7, allowing the formation of a zinc-bound hydroxide ion at physiological pH[26]. The zinc ion can be substituted by other physiologically relevant transition metal ions such as $Co^{2+}$, $Ni^{2+}$, $Cu^{2+}$, $Cd^{2+}$, and $Mn^{2+}$ which results in drastic changes in the catalytic activity of CA II (~50% active to completely inactive)[21]. It has been also reported that the metal substitutions may induce alternative catalytic activities of CA II other than $CO_2/HCO_3^-$ conversion[27], for instance, reduction of nitrite to nitric oxide in presence of copper[28].

Previous structural studies had suggested that different metal coordination geometries in the non-native CA II may play an important role in their catalysis[29,30], but no clear evidence was presented to support such a claim. Our present study focuses on investigating the detailed structural changes in CA II during the $CO_2/HCO_3^-$ conversion catalysis and correlating these variations to the relevant catalytic mechanisms. These experimental insights

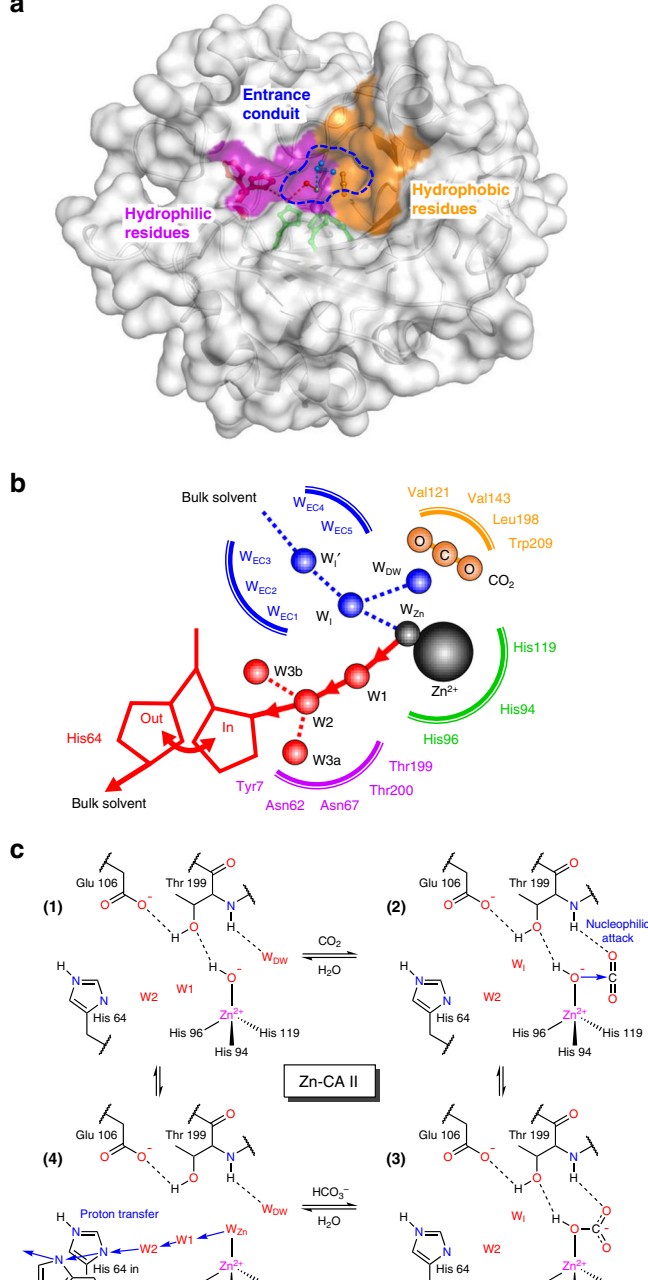

**Fig. 1 Structure of native carbonic anhydrase II (Zn-CA II) and its catalytic mechanism. a** The active site consists of zinc binding site, hydrophobic/hydrophilic regions, and entrance conduit (EC). **b** The water networks in the active site are responsible for the proton transfer (red) and substrate/product/water exchange (blue) during enzyme catalysis. **c** The $CO_2$ hydration reaction mechanism of Zn-CA II. First, $CO_2$ binds to the active site, leading to a nucleophilic attack by the zinc-bound hydroxyl ion onto $CO_2$. $HCO_3^-$ thus formed is subsequently displaced by the water molecule inflowing through EC. The $HCO_3^-$ molecule likely binds to $Zn^{2+}$ ion in a monodentate mode and its OH group is held at the $Zn^{2+}$ ion due to the hydrogen bonding with Thr199[52, 53]. This product binding configuration leads to a weak interaction between the product and $Zn^{2+}$ ion, thereby facilitating fast product dissociation[54]. Finally, proton transfer occurs via the network ($W_{Zn} \rightarrow W1 \rightarrow W2 \rightarrow$ His64) provided by the protein scaffold.

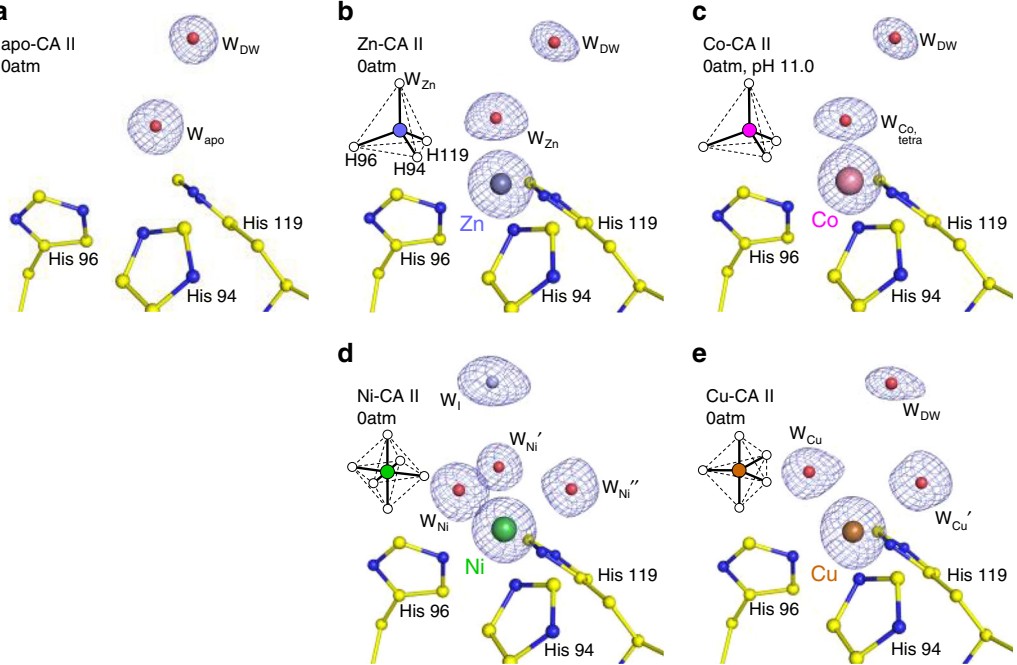

**Fig. 2 Metal coordination geometry in CA II without CO₂ pressurization. a** In apo-CA II, the metal binding site is vacant. **b, c** Zn- and Co-CA II show tetrahedral, **d** Ni-CA II octahedral, and **e** Cu-CA II trigonal bipyramidal coordination geometry. The electron density (2F$_o$–F$_c$, blue) is contoured at 2.2σ. All structures were obtained at pH 7.8 except for (**c**) which is obtained at pH 11.0. The intermediate water (W$_I$) in (**d**) is colored in steel blue for clarity.

offer us a fresh peek into the origin of the activity alterations caused by non-native metal substitutions.

To study the role of metal ions in CA II, we selected four divalent transition-metal ions ($Zn^{2+}$, $Co^{2+}$, $Ni^{2+}$, and $Cu^{2+}$) that induce drastic changes in CA II activity (100%, ~50%, ~2%, and 0%, respectively)[31,32]. The catalytic intermediate states of the metal-free (apo, as a control) and the four metal-bound CA IIs were prepared by cryocooling protein crystals under $CO_2$ pressures from 0 (no $CO_2$ pressurization) to 20 atm[33,34].

We show that the characteristic metal ion coordination geometries directly modulate the catalytic processes, including substrate binding, its conversion to product, and product binding. In addition, we reveal that the metal ions have a long-range (~10 Å) electrostatic effect on restructuring the water network at the active site, affecting the product displacement and the proton transfer process. The cumulative effect of such alterations provides mechanistic insights into the overall reduction of the enzymatic activity in the non-native metal-substituted CA IIs.

## Results

**The role of metal ion coordination geometries.** The coordination geometry around the metal binding site in CA II, when no $CO_2$ pressure is applied, is shown in Fig. 2. The metal-free apo-CA II shows an electron density map reflecting the presence of a water molecule in the metal binding site (Fig. 2a). In Zn-CA II and Co-CA II (pH 11.0), the metal ions display tetrahedral coordination with three histidine residues (His94, His96, and His119) and a water molecule (Fig. 2b, c). In contrast, Ni-CA II contains three bound water molecules, completing an octahedral (hexa-coordinate) geometry (Fig. 2d). Finally, Cu-CA II possesses two bound water molecules, arranged in trigonal bipyramidal (penta-coordinate) geometry (Fig. 2e).

Next, we investigated the effect of metal coordination geometry on the efficacy of substrate ($CO_2$) and product ($HCO_3^-$) binding. The apo- and Zn-CA II structures cryocooled at 20 atm $CO_2$ pressure are shown in Fig. 3. The apo-CA II shows a clear binding

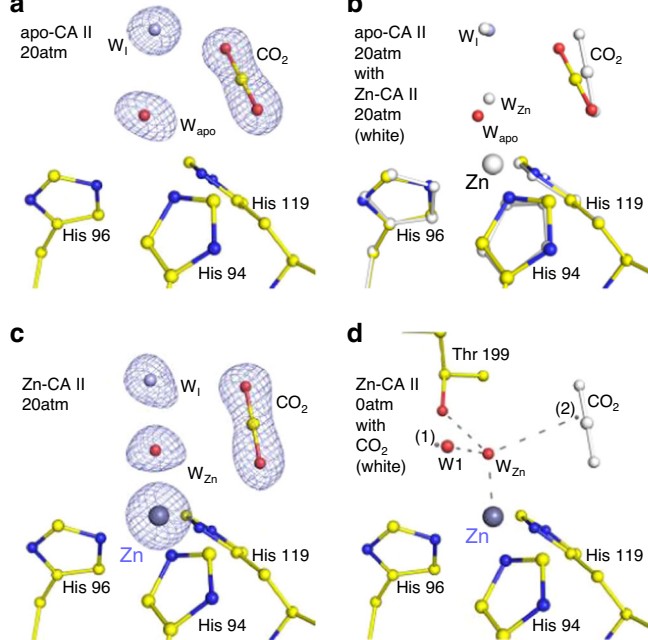

**Fig. 3 Substrate/product binding in apo- and Zn-CA II.** The intermediate water (W$_I$) is colored in steel blue for clarity. The electron density (2F$_o$–F$_c$, blue) is contoured at 2.2σ. **a, b** At 20 atm of $CO_2$ pressure, apo-CA II shows clear binding of $CO_2$ without the need of $Zn^{2+}$ ion. **c** Zn-CA II shows similar binding of $CO_2$ as in apo-CA II while maintaining tetrahedral metal coordination. **d** Upon $CO_2$ binding (white) in Zn-CA II, W$_{Zn}$ is located at the center of the hypothetical tetrahedral arrangement made up of $Zn^{2+}$ ion, Thr199-Oγ1, position (1) (close to W1), and position (2) (close to the carbon atom in $CO_2$). In this configuration, a hybridized lone pair in W$_{Zn}$ directly faces $CO_2$ molecule at a distance, appropriate for efficient nucleophilic attack. Distance between the position (2) and C atom of $CO_2$ is merely 0.36 Å.

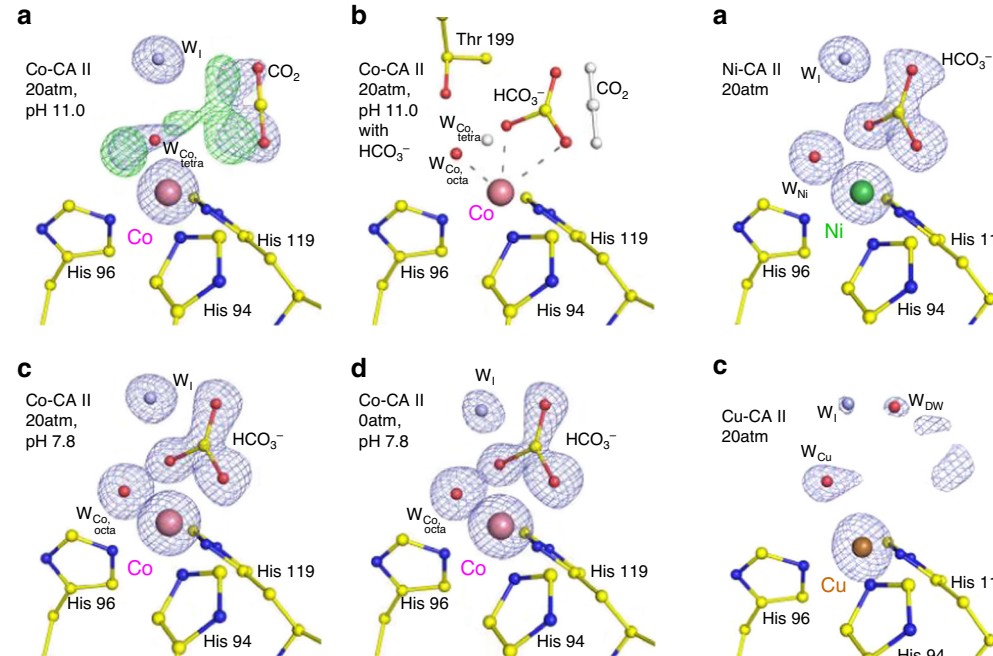

**Fig. 4 Substrate/product binding in Co-CA II.** The intermediate water ($W_I$) is colored in steel blue for clarity. The electron density ($2F_o$–$F_c$, blue) and the difference map ($F_o$–$F_c$, green) are contoured at $2.2\sigma$ and $7.0\sigma$, respectively. **a**, **b** At 20 atm of $CO_2$ pressure, Co-CA II at pH 11.0 shows superposition of $CO_2$ binding (~50% occupancy, white) with tetrahedral coordination and $HCO_3^-$ binding (~50% occupancy) with octahedral coordination. **c**, **d** Co-CA II at pH 7.8 shows complete binding of $HCO_3^-$, showing octahedral coordination even in absence of added $CO_2$. It is likely that the captured $HCO_3^-$ is converted from the $CO_2$ absorbed in the crystal from ambient air.

**Fig. 5 Substrate/product binding in Ni- and Cu-CA II.** The intermediate water ($W_I$) is colored in steel blue for clarity. The electron density ($2F_o$–$F_c$, blue) is contoured at $2.2\sigma$. **a** At 20 atm of $CO_2$ pressure, Ni-CA II maintains octahedral coordination with $HCO_3^-$ binding. **b** Compared to the $W_{Zn}$ geometry in Zn-CA II (Fig. 3d), the nucleophilic attack geometry around $W_{Ni}'$ has steric hindrance on $CO_2$ molecule (adapted from Zn-CA II, 20 atm, white) and is distorted away. Distance between the position (2) and C atom of $CO_2$ is 1.55 Å. **c** Cu-CA II shows only disordered electron density in the $CO_2/HCO_3^-$ binding site. **d** The nucleophilic attack geometry around $W_{Cu}$ has steric hindrance on $CO_2$ molecule (adapted from Zn-CA II, 20 atm) and is significantly distorted away. Distance between the position (2) and C atom of $CO_2$ is 2.93 Å.

of $CO_2$ molecule, replacing deep water, $W_{DW}$ present within the active site, suggesting that the $CO_2$ binding at the active site is mostly dictated by the metal-free protein scaffold (Fig. 3a). The native holoenzyme Zn-CA II shows $CO_2$ binding almost identical to that in apo-CA II (Fig. 3b, c). The $CO_2$ molecule is located 2.9 Å away from the Zn-bound water ($W_{Zn}$), in a configuration conducive for the nucleophilic attack (Fig. 3d).

In Co-CA II (pH 11.0) cryocooled at 20 atm $CO_2$ pressure, dual binding of $CO_2$ and $HCO_3^-$ is observed (Fig. 4a). Upon $CO_2$ binding, the tetrahedral coordination is maintained, but an unusual expansion to octahedral coordination is observed upon $HCO_3^-$ binding (Fig. 4b). In the transformed octahedral geometry, the $HCO_3^-$ molecule is bound in a bidentate mode to the $Co^{2+}$ ion along with an additional water molecule. Compared to the monodentate binding mode in Zn-CA II, the negative charge on the bidentate $HCO_3^-$ can be distributed among the two oxygen atoms bound to $Co^{2+}$ ion, allowing stronger product binding to the metal ion (Supplementary Fig. 1a–f). Unlike Zn-CA II (Supplementary Fig. 2), the Co-CA II intermediates obtained at different pH values (7.8 and 11.0) reveal that the $HCO_3^-$ molecule is firmly bound to $Co^{2+}$ ion with full occupancy at lower pH (Fig. 4c, d), but this binding affinity weakens as pH increases (Fig. 4a). The result suggests that, during the catalytic cycle, deprotonation of the $Co^{2+}$-bound water may lead to dissociation of the $HCO_3^-$ molecule from the $Co^{2+}$ ion, due to the charge–charge repulsion between the formed hydroxide ion and the $HCO_3^-$ molecule. Following the $HCO_3^-$ dissociation, the tetrahedral coordination is restored (Fig. 2c).

On the other hand, at 20 atm $CO_2$ pressure, Ni-CA II shows octahedral coordination comprising the bidentate $HCO_3^-$ and a

water molecule in a similar manner to that of Co-CA II (Fig. 5a, Supplementary Fig. 1g–i). It is noted that one of the three bound water molecules experiences steric hindrance with the $CO_2$-binding configuration in Zn-CA II (Fig. 5b). Thus, it is likely that the $CO_2$ molecule entering the active site pushes away one of the Ni-bound water molecules, and then a nucleophilic attack occurs from one of the two remaining water molecules, forming $HCO_3^-$. Unlike Co-CA II (Fig. 4a, c, d), the Ni-CA II intermediates obtained at different pH values (7.8 and 11.0) indicate that the $HCO_3^-$ binding affinity is almost unresponsive to pH variation (Supplementary Fig. 3). The result suggests that the deprotonation of the $Ni^{2+}$-bound water is insufficient to facilitate $HCO_3^-$ dissociation in the stable octahedral coordination, and that the bound $HCO_3^-$ is directly displaced by two incoming water molecules in Ni-CA II. Finally, in Cu-CA II, no clear electron density of $CO_2$ or $HCO_3^-$ is visible (Fig. 5c). The faint and diffused electron density suggests that a $CO_2$ molecule encounters a severe steric hindrance from one of $Cu^{2+}$-bound water molecules. Even if the $CO_2$ molecule adopts proper orientation as in Zn-CA II, the bound $CO_2$ position remains far too distant (3.9 Å) from the spare $Cu^{2+}$-bound water molecule for any effective interaction. Moreover, the significantly distorted geometry negates the scope of any nucleophilic attack (Fig. 5d). The inefficient substrate binding and the unfavorable distorted geometry explain the complete enzymatic inactivity of Cu-CA II.

**Electrostatic effects of metal ions on active-site water network.** Figure 6 shows the proton transfer pathway and the water

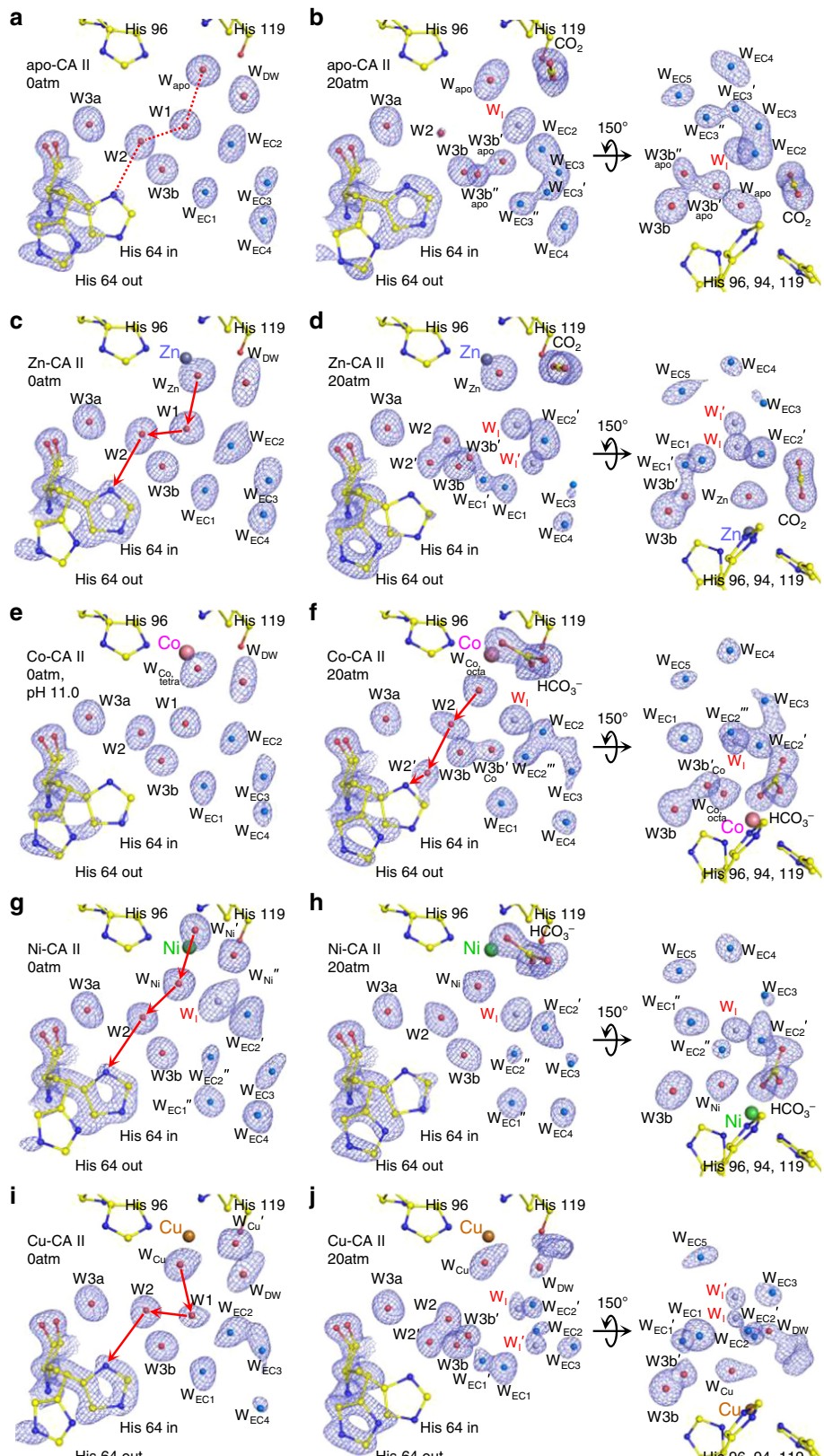

network in the EC of CA II. The metal-free apo-CA II shows the well-defined pathway ($W_{Apo} \rightarrow W1 \rightarrow W2 \rightarrow His64$, Fig. 6a) in the absence of $CO_2$. Upon $CO_2$ binding, the pathway is disrupted in such a way that W1 disappears and an intermediate water $W_I$ emerges (Figs. 3a and 6b). Zn-CA II largely resembles the apoenzyme in terms of both the same well-defined pathway

($W_{Zn} \rightarrow W1 \rightarrow W2 \rightarrow His64$) utilized for proton transfer (Fig. 6c) and the dynamics of $W1/W_I$ upon $CO_2$ binding (Figs. 3c and 6d). This observation clearly suggests that the primary water network necessary for the proton transfer is organized by the protein scaffold without the need for metal ions. However, in comparison to apo-CA II (Fig. 6b), Zn-CA II shows significantly

**Fig. 6 Active site in CA II showing proton transfer pathway and EC water network ($W_{EC1} \sim W_{EC5}$).** The electron density ($2F_o–F_c$) is contoured at $1.7\sigma$ except for EC waters at $1.5\sigma$. The EC waters are colored in aqua marine and the intermediate waters ($W_I$ and $W_I'$) in steel blue for clarity. W2' is an alternative position of W2. The possible proton transfer pathways in the metal-CA IIs are depicted as red arrows. All structures were obtained at pH 7.8 except for **e** at pH 11.0. **a**, **b** Apo-CA II shows well-ordered water arrangement (dotted red line) with His64 favored in outward conformation at 0 atm $CO_2$ pressure. Upon $CO_2$ binding, His64 moves inward and water molecules show highly dynamical motions, stabilizing $W_I$. **c**, **d** Zn-CA II shows His64 favored in inward conformation at 0 atm $CO_2$ pressure. Upon $CO_2$ binding, W2, W3b, and $W_{EC}$ waters show significantly different dynamics with His64 moving outward, and an additional intermediate water ($W_I'$) is stabilized with the $W_{EC}$ molecules. The motions of W3b and $W_{EC1}$ turn on the dynamic interplay between the proton transfer and EC water networks. **e**, **f** Co-CA II shows similar arrangement initially as in Zn-CA II. However, upon full $HCO_3^-$ binding, the dynamical motions of EC waters are different and the intermediate water $W_I'$ is less stabilized. Note that, in Co-CA II, proton transfer seems to occur while the product is still bound. **g**, **h** Ni-CA II initially shows altered water arrangements due to octahedral coordination. Upon $HCO_3^-$ binding, significantly reduced water dynamical motions are recognized. **i**, **j** Cu-CA II shows unexpectedly similar dynamical motions of active site waters and His64 as in Zn-CA II.

modified dynamics of W2 and His64, which are believed to be critical for efficient proton transfer (Fig. 6d). Additionally, in Zn-CA II, the significantly modified dynamics of the EC waters stabilizes another intermediate water $W_I'$, that in turn bridges $W_I$ with the bulk solvent outside the protein, thereby facilitating the replenishment of $W_{Zn}$ and W1 during the catalytic cycle (Figs. 1b and 6d). These observations suggest that the $Zn^{2+}$ ion produces a long-range (~10 Å) electrostatic field in which water structure and dynamics in the active site are fine-tuned to facilitate the proton transfer and the water/substrate/product exchange.

In the absence of $CO_2$, Co-CA II forms the same proton transfer network as in Zn-CA II (Fig. 6e). Once $CO_2$ or $HCO_3^-$ binds, W1 disappears and $W_I$ appears like what happens in Zn-CA II (Figs. 4a, c, and 6f). However, as the deprotonation of $Co^{2+}$-bound water should occur prior to the $HCO_3^-$ dissociation, it is likely that the proton transfer occurs via the altered network ($W_{Co,octa} \rightarrow W2 \rightarrow His64$) while the product is still bound to the $Co^{2+}$ ion (Fig. 6f). In addition, Co-CA II shows modified dynamics of W2, His64, and EC waters as compared to Zn-CA II (Fig. 6f). Meanwhile, in Ni-CA II, octahedral coordination is stabilized throughout the entire catalytic cycle, and consequently, W1 is absent due to its steric hindrance with one of the Ni-bound water molecules (Figs. 2d, 5a, and 6g). Based on the $CO_2$ binding configuration in Zn-CA II and the bidentate $HCO_3^-$ binding observed in Ni-CA II, it is most likely that the substrate-to-product conversion occurs via the nucleophilic attack from $W_{Ni}'$ to $CO_2$ (Fig. 5b). This result suggests that the proton transfer occurs possibly via the modified pathway $W_{Ni}' \rightarrow W_{Ni} \rightarrow W2 \rightarrow His64$ (Fig. 6g). Also, unlike Zn-CA II and Co-CA II, W2 in Ni-CA II shows significantly different dynamics and $W_I'$ is destabilized, a plausible reflection of the altered electrostatic environment (Fig. 6h). Finally, Cu-CA II reveals that the possible proton transfer pathway ($W_{Cu} \rightarrow W1 \rightarrow W2 \rightarrow His64$) is well-defined (Fig. 6i) and W2 and His64 dynamics is surprisingly similar to that in Zn-CA II (Fig. 6j). This result corroborates well with our conjecture that it is the lack of efficient substrate binding and unfavorable distorted geometry for the nucleophilic attack that are responsible for the complete inactivity of Cu-CA II.

## Discussion

Our results provide advanced insights into the role of metal ions and the metal-protein relationship for the CA II catalytic mechanism. In the absence of metal ions, the protein scaffold provides a fundamental structural template necessary for the catalytic activity. The protein scaffold helps usher a substrate molecule from the outside bulk solvent into the active site through desolvating and positioning it at a configuration conducive for nucleophilic attack. The protein scaffold also provides well-ordered water networks in the vicinity of the active site, which can be utilized for proton transfer and substrate/product/water exchange. Metal ions then bring a key property for the CA II catalytic activity in generating hydroxyl ion at neutral pH and

retaining it at the active site. Beyond their primary Lewis acid property, metal ions are directly involved in the catalytic mechanism via their coordination geometry and long-range electrostatic effects. The most efficient native Zn-CA II preserves a tetrahedral coordination and fine-tunes the water network embedded within the protein scaffold (Fig. 1c). The tetrahedral coordination allows efficient conversion of substrate into product, and the long-range electrostatic field orchestrates the structure and dynamics of water network in the active site, imperative for the rapid product displacement and fast proton transfer. In comparison, semi-efficient Co-CA II shows similar catalytic behavior up to the product formation stage as in Zn-CA II, but the expansion of the metal coordination geometry from tetra-hedron to octahedron during the catalytic cycle alters the product displacement and proton transfer process (Fig. 7). The significantly less efficient Ni-CA II maintains octahedral coordination and shows altered electrostatic effects, hampering efficient conversion from substrate to product, product displacement, and proton transfer (Fig. 8). Finally, completely inactive Cu-CA II suggests substantial steric hindrance encountered by the substrate in the active site and poor geometry for product conversion due to the trigonal bipyramidal coordination (Fig. 9).

In conclusion, we examined the role of various metal ions in carbonic anhydrase catalysis beyond their primary chemical property as a Lewis acid. We demonstrated that metal ions are directly involved in the enzymatic mechanism via their coordination geometry and long-range electrostatics to orchestrate intricate water dynamics. Our experimental results can be used as direct input for theoretical and computational studies on the role of metal ions, which we anticipate could open a new window to the study of metal–protein relationships, drug discovery targeting metalloenzymes, engineering of natural metalloenzymes, rational design of de novo metalloenzymes, and synthesis of supramolecular analogues to metalloenzymes.

## Methods

**Protein expression and purification**. The native Zn-CA II was expressed in a recombinant strain of *Escherichia coli* [BL21 (DE3) pLysS] containing a plasmid encoding the CA II gene[35]. Purification was carried out using affinity chromatography[36]. Briefly, bacterial cells were enzymatically lysed with hen egg white lysozyme, and the lysate was placed onto an agarose resin coupled with p-(ami-nomethyl)-benzene-sulfonamide which binds CA II. The protein on the resin was eluted with 0.4 M sodium azide, in 100 mM Tris-HCl pH 7.0. The azide was removed by extensive buffer exchange against 10 mM Tris-HCl pH 8.0.

Apo-CA II (zinc free) was then prepared by incubating Zn-CA II in a zinc chelation buffer (100 mM pyridine 2,6-dicarboxylic acid, 25 mM MOPS pH 7.0) at 20 °C for 18 h. The resulting protein was then run through an affinity column with benzylsulfonamide resin to remove residual Zn-CA II. The chelating agent was then removed by buffer-exchange against 50 mM Tris-HCl pH 7.8[18]. The loss of zinc ion was examined using the esterase kinetic assay and further confirmed in the crystallographic structure. The enzyme activity was revived by an addition of 1 mM $ZnCl_2$.

**Esterase kinetic assay**. The $CO_2/HCO_3^-$ conversion catalytic activity of CA II can be measured directly by stopped flow assays, monitoring labeled $CO_2/HCO_3^-$

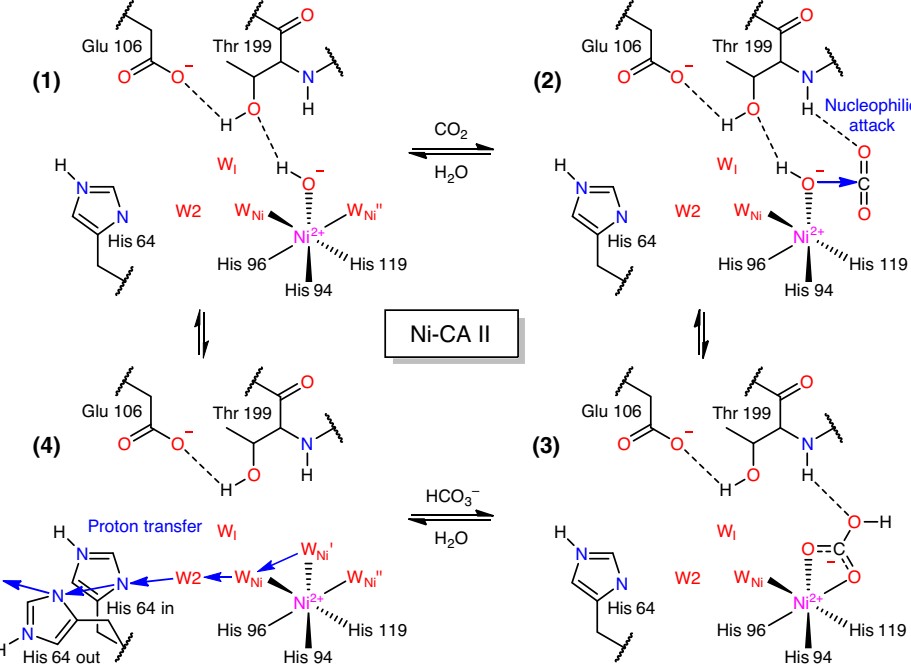

**Fig. 7 Proposed catalytic mechanism of Co-CA II.** In Co-CA II, $CO_2$ binding and the catalytic conversion of $CO_2$ to $HCO_3^-$ occur in the same way as in the Zn-CA II with tetrahedral geometry. However, the $HCO_3^-$ displacement and proton transfer process are significantly altered due to the coordination expansion to octahedral geometry during catalysis. This octahedral coordination allows bidentate binding mode of $HCO_3^-$ and reorganization of negative charge of $HCO_3^-$ toward $Co^{2+}$ ion, allowing stronger $HCO_3^-$ binding to metal ion. To dissociate the product, proton transfer first occurs via an altered pathway (possibly, $W_{Co,octa} \rightarrow W2 \rightarrow His64$) and $W_{Co,octa}$ is converted into the hydroxyl ion. This negatively charged hydroxyl ion then pushes away the bound product, and the tetrahedral coordination is restored for the next catalytic cycle.

**Fig. 8 Proposed catalytic mechanism of Ni-CA II.** In Ni-CA II, octahedral coordination is maintained throughout the whole catalytic cycle. The significant consequence is that one of the three bound water molecules experiences steric hindrance with the $CO_2$ binding. In addition, the nucleophilic attack geometry is distorted (Fig. 5b), suggesting less efficient conversion into $HCO_3^-$. The formed $HCO_3^-$ is strongly bound to $Ni^{2+}$ ion in a bidentate mode as in the Co-CA II but is directly displaced by two inflowing water molecules. Finally, proton transfer occurs via an altered network (possibly, $W_{Ni}' \rightarrow W_{Ni} \rightarrow W2 \rightarrow His64$) to restore the catalytic cycle.

**Fig. 9 Proposed catalytic mechanism of Cu-CA II.** In Cu-CA II, one of the two bound water molecules in the trigonal bipyramid coordination experiences significant steric hindrance from $CO_2$ molecule, hindering adoption of proper configuration for nucleophilic attack. In addition, even if $CO_2$ binds to the active site temporarily, the nucleophilic attack geometry is too distant (3.9 Å) and significantly distorted (Fig. 5d).

conversion using mass spectroscopy, or indirectly by monitoring the innate esterase activity spectroscopically[37,38]. In this study, the esterase activity assays were performed as a control to ensure zinc was fully chelated from recombinant CA II. The 4-nitrophenyl acetate (4-NPA) molecule is cleavable by CA II and thus used here as a colorimetric substrate. CA II cleaves the ester bond of 4-NPA generating 4-nitrophenol, which is spectroscopically absorbent at 348 nm in the ultraviolet–visible spectrum. Thus, the reaction can be monitored spectroscopically at 348 nm[39].

In a 96 deep-well plate, aliquots of 50 μL of 0.1 mg mL$^{-1}$ CA II in storage buffer were added to each well. To initiate the reaction, 200 μL of 0.8 mM 4-NPA dissolved in 3% acetone in water was added to the sample well. The well plate was then immediately inserted into the plate reader (Synergy HTX, BioTek, Winooski, WI, USA). Absorbance at 348 nm was recorded every 8 s for 10 min. The absorbance data of Apo- and Zn-CA II are plotted in Supplementary Fig. 4.

**Crystallization and non-native metal substitution.** Crystals of CA II were obtained using the hanging drop vapor diffusion method[40]. A 10 μl drop of equal volumes of protein (5 μl) and the well-solution (5 μl) was equilibrated against 500 μl of the well-solution (1.3 M sodium citrate, 50 mM Tris-HCl pH 7.8) at RT (~20 °C)[41]. Crystals grew to an approximate ~30 × 100 × 200 μm$^3$ in size in a few days. To prepare non-native metal substituted CA II, the apo-CA II crystals were transferred into soaking solutions of cobalt, nickel and copper salt (100 mM CoCl$_2$, 100 mM NiCl$_2$, 10 mM CuCl$_2$ along with 1.3 M sodium citrate, 50 mM Tris-HCl with pH 7.8). The crystals were incubated for 2–3 days to let the $Co^{2+}$, $Ni^{2+}$ and $Cu^{2+}$ ions infuse into the active site[42]. The CA II crystals at pH 11.0 were obtained with 3-(cyclohexylamino)propanesulfonic acid buffer instead of using Tris-HCl.

**Cryocooling under $CO_2$ pressure.** Cryo-trapping the intermediate states of Zn-CA II was previously achieved by cryocooling CA II crystals under $CO_2$ pressure[33,34], leading to the capture of $CO_2$ in the active site of CA II[43]. More recently, series of intermediate states have been tracked in CA II by controlling the internal $CO_2$ pressure levels[25,44]. In this study, the $CO_2$ entrapment was carried out using a high-pressure cryo-cooler for X-ray crystallography (HPC-201, Advanced Design Consulting, USA). The apo-, Zn-, Co-, Ni-, and Cu-CA II crystals were first soaked in a cryo-solution containing 35% (v/v) glycerol supplemented to the soaking solution. The crystals were then coated with mineral oil to prevent dehydration, and loaded into the base of high-pressure tubes[33]. The coated mineral oil worked as a $CO_2$ buffering medium as well, aiding in the absorption of $CO_2$ into the crystals[45]. The crystals were pressurized at room temperature in the pressure tubes with $CO_2$ gas at 0 atm (no pressurization) and 20 atm. After a wait of about 5 min, the crystals were cryocooled in liquid nitrogen (77 K). Once the $CO_2$ bound crystals were fully cryocooled, the $CO_2$ gas pressure was withdrawn, and the crystal samples were stored in a liquid nitrogen dewar for subsequent X-ray data collection.

**X-ray diffraction and data collection.** Diffraction data were collected at Pohang Light Source II (wavelength of 0.9793 Å, beam size of 100 μm) under nitrogen cold stream (100 K). Data were collected using the oscillation method in intervals of 1° step on an ADSC Quantum 270 CCD detector (Area Detector Systems Corporation, USA) with a crystal-to-detector distance of 120 mm. A total of 360 images were collected on each of the CA II crystal data sets.

For each data set, a new fresh pressure-cryocooled crystal was used. The absorbed X-ray dose for a single data set was less than $5 × 10^5$ Gy, which is much less than the Henderson dose limit of $1.2 × 10^7$ Gy[46]. Moreover, we have checked that X-ray radiation dose at least up to $10^7$ Gy does not induce apparent changes in the active site. The result confirms that the active site structures described in our study are unaffected by the X-ray radiation. Indexing, integration, and scaling were performed by using HKL2000[47]. The data processing statistics are given in Supplementary Table 1.

**Structure determination and model refinement.** The CA II structures were determined using the CCP4 program suite[48]. Prior to refinement, a random 5% of the data were flagged for $R_{free}$ analysis. The previously reported crystal structures (PDB codes of 5DSR and 5YUK for apo- and metal substituted CA II) were used as the initial phasing models[25,49]. The maximum likelihood refinement (MLH) was carried out using REFMAC5[50]. The refined structures were manually checked using the molecular graphics program COOT[51]. Reiterations of MLH were carried out with anisotropic B factor.

On completion of the structural refinements as described above, systematic refinements were further carried out to accurately determine the partial occupancies of the His 64 in and the His 64 out configurations. A total of 99 structures were prepared for each of the CA II structures, in which the occupancies of the His 64 in and the His 64 out configurations were changed in incremental steps of 1% (i.e., the first structure with 1% in and 99% out, the second structure with 2% in and 98% out, …, the 99th structure with 99% in and 1% out). MLH refinements were carried out in parallel for all the 99 structures. After MLH refinements, the overall $R$-factor as a function of partial occupancy of the His 64 in configuration was obtained, and it was fitted into a quadratic function (Supplementary Fig. 5). The partial occupancy values of the His 64 configurations were determined where the overall R-factor is minimized. Details on the final refinement statistics are given in Supplementary Table 1. All structural figures were rendered with PyMol (Schrödinger, LLC).

**Structural analysis of the bound water molecules.** To compare the bound water molecules in the active site and the EC, we carefully refined water molecules based on the PDB and COOT validation checks and the electron density maps (cutoff level of $1\sigma$ in $2F_o$–$F_c$ electron density map). We have tested the consistency and reproducibility of the bound water molecules in the active site and the EC carefully. There were several closely positioned water molecules in the active site and the EC of the CA II structures. Since most of these waters exist transiently, it was allowed that they can be located closer than the normal stably bound water molecules. In this regard, water molecules closely located near the active site and EC regions were not excluded in the final coordinates. The important bound water molecules addressed in the main paper are listed in Supplementary Table 2. The distance information between $CO_2$, $HCO_3^-$, Thr199, and important water molecules is listed in Supplementary Table 3.

**Reporting summary.** Further information on research design is available in the Nature Research Reporting Summary linked to this article.

## Data availability

The atomic coordinates and structure factors have been deposited in the Protein Data Bank (http://wwpdb.org/) as [PDB code 6LUU [https://doi.org/10.2210/pdb6luu/pdb] (0 atm $CO_2$ pressure, pH 7.8), 6LUV [https://doi.org/10.2210/pdb6luv/pdb] (20 atm, pH 7.8)] for apo-CA II, [6LUW [https://doi.org/10.2210/pdb6luw/pdb] (0 atm, pH 7.8), 6LUX [https://doi.org/10.2210/pdb6lux/pdb] (20 atm, pH 7.8), 6LUY [https://doi.org/10.2210/pdb6luy/pdb] (0 atm, pH 11.0), 6LUZ [https://doi.org/10.2210/pdb6luz/pdb] (20 atm, pH 11.0)] for Zn-CA II, [6LV1 [https://doi.org/10.2210/pdb6lv1/pdb] (0 atm, pH 7.8), 6LV2 [https://doi.org/10.2210/pdb6lv2/pdb] (20 atm, pH 7.8), 6LV3 [https://doi.org/10.2210/pdb6lv3/pdb] (0 atm, pH 11.0), 6LV4 [https://doi.org/10.2210/pdb6lv4/pdb] (20 atm, pH 11.0)] for Co-CA II, [6LV5 [https://doi.org/10.2210/pdb6lv5/pdb] (0 atm, pH 7.8), 6LV6 [https://doi.org/10.2210/pdb6lv6/pdb] (20 atm, pH 7.8), 6LV7 [https://doi.org/10.2210/pdb6lv7/pdb] (0 atm, pH 11.0), 6LV8 [https://doi.org/10.2210/pdb6lv8/pdb] (20 atm, pH 11.0)] for Ni-CA II, and [6LV9 [https://doi.org/10.2210/pdb6lv9/pdb] (0 atm, pH 7.8), 6LVA [https://doi.org/10.2210/pdb6lva/pdb] (20 atm, pH 7.8)] for Cu-CA II. Two earlier structures [5DSR [https://doi.org/10.2210/pdb5dsr/pdb] and 5YUK

[https://doi.org/10.2210/pdb5yuk/pdb]] were used for structure determination. Source data are provided with this paper.

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

## Acknowledgements

The authors would like to thank the staff at Pohang Light Source II for their support in data collection. This work was initiated by the support of Samsung Science and Technology Foundation (SSTF-BA1702-04) and further supported by the National Research Foundation of Korea (NRF) grant (NRF-2019R1A2C1004274) funded by the Korea government (MSIT).

## Author contributions

C.U.K. conceived the research, J.K.K., C.L., S.W.L., J.T.A. ran the experiments, J.K.K. and C.U.K. analysed the data. J.K.K., A.A., R.M., C.-M.G., and C.U.K. wrote the paper. All authors contributed to the overall scientific interpretation and edited the paper.

## Competing interests

The authors declare no competing interests.

**Additional information**

**Peer review information** *Nature Communications* thanks Ian Davis and the other, anonymous reviewer(s) for their contribution to the peer review of this week. Peer review reports are available.

