## [Peer Review File · Nature Communications]

REVIEWER COMMENTS

Reviewer #1 (Remarks to the Author):

The manuscript by J. K. Kim et al. provides many new crystal structures of different metal variants of human carbonic anhydrase II with and without a substrate molecule captured in its catalytic center. Specific knowledge and techniques were employed since the catalyzed reaction of the analyzed enzyme is very fast and the substrate is gaseous. The authors analyzed metal coordination geometry, its effect on catalytic activity and water molecule network. The results will be interesting and useful to the carbonic anhydrase research field, but also could be used for better understanding of other metalloenzyme systems. I believe the information could be employed for more precise computer simulations/ docking procedures of small molecule inhibitors for metalloenzymes.

The manuscript is clearly written, with adequate citation and optimal manuscript length. I would suggest adding a short discussion about a possible alternative catalytic activity that may emerge for different metallovariants of the enzyme, as was recently reported by the same group (DOI: 10.1107/S2052252520000986) for copper-CA II.

Some comments: authors state (lines 18-21): „ Through a comparative study on the intermediate states of the zinc-bound native human CA II and non-native metal-substituted CA IIs, we demonstrate that the characteristic metal ion coordination geometries <...> directly modulate the catalytic efficacy”. Then the lines 66-68 reference to Krishnamurthy et al. 2008 review for results on percentage of catalytic activity of different metal variants CA (would it be more precise if a primary source for these results was indicated?). In methods section lines 195-197 state “The loss of enzyme activity was verified using kinetic studies. The enzyme activity was revived by an addition of 1 mM ZnCl₂.” However, no experimental data on enzymatic activity are provided in this manuscript. Could the authors describe the method used to quantify enzymatic activity and provide raw data of the enzymatic activity in the Supplementary material?

In my opinion, this manuscript is of high scientific quality and could be published in Nature communications if the requested information about the catalytic activity is provided.

Reviewer #2 (Remarks to the Author):

The paper by Kim et al. reports a detailed study by means of X-ray crystallography on the effect of metal ion substitution in the carbonic anhydrase II active site, showing that the change of coordination geometry modulates the catalytic efficiency and the network of water molecules in the enzyme active site.

The paper is rather interesting, technically sound and logically arranged; however, in my opinion it does not possess sufficient novelty to be published in a so high impact factor journal such as Nature Communication.

Reviewer #3 (Remarks to the Author):

This work seeks to systematically evaluate the contribution of the catalytic metal of human carbonic anhydrase II using X-ray crystallography. Sixteen high-resolution structures were obtained under various conditions from apo-enzyme to Zn, Co, Ni, or Cu loaded with and without CO₂ pressure at pH 7.8 or 11. This manuscript represents a substantial undertaking which provides high-quality, information-dense data sets for the metallobiochemistry field. My major concern is that the compactness of the presentation may lessen impact for readers who are not already intimately familiar with the CA II system. Also, some conclusions (detailed below) are drawn too definitely considering that this study does not include any biochemical or enzymological

characterizations.

Major Concerns:

Figure 2 condenses too much information and too many ideas. It contains information about i) changes in coordination geometry upon metal binding/substitution (a-e), ii) geometry of CO₂ binding (f,g,h), iii) geometry of HCO₃ binding (i,m), iv) how pH affects CO₂ vs HCO₃ binding (h,m), v) effect of Zn on CO₂ binding (k), and iv) steric clashes of hypothetical CO₂ binding (n,o). While this is certainly space efficient, I think there is enough interesting data here to justify 2 or 3 separate figures with more narrow focus. If this advice is taken, consider moving panels from Supplementary figure 3 to the main text.

Page 5, lines 89-95: The authors note that carbonate binds Co-CA II in a bidentate fashion, leading to a higher binding affinity which in turn reduces overall activity. While bidentate rather than monodentate binding may correlate with a higher binding affinity, it is not sufficient to draw such a conclusion, especially regarding the solution state behavior of an enzyme.

Supplementary Fig. 1c&f: Unless proton positions have been experimentally determined (Neutron diffraction or spectroscopically) or supported by computational efforts, it is best to leave their positions unassigned.

Page 6, line 117: Avoid using "This" as the subject of a sentence. Also, adding a qualifier to this sentence seems appropriate.

The second and third column of Figure 3 appear to show different angles of the same data set. Such presentation is usually accompanied by a curved arrow labelling the degree of rotation. In this case, following convention would also reduce the number of panel labels needed by 5. Also, the degree to which the conformation of His64 is altered does not seem significant enough to warrant inclusion in the figure. These values may be better presented in a table in the SI.

Minor Concerns:

The abstract mentions "unprecedented roles of metal ions in a model metalloenzyme system," however I did not find anything unprecedented about the roles of the metal ions in this work.

Fig. 1c is not referenced until the discussion. It should be discussed before Fig 2 or moved to a later figure.

Page 5, lines 78 & 88: It should be mentioned that the Co-CA II structure (Fig. 2c) was obtained under a different condition (pH 11.0) than all other structures (pH 7.8).

Page 6, lines 108,9: Deprotonation of the metal-bound water is proposed as cause of carbonate dissociation in the case of Co. Can the authors provide any insight as to why that same reasoning cannot be applied to Ni?

Page 7, line 131: If it is a fact that Zn produces a long-range electrostatic field, then an appropriate citation should be provided.

Page 7, line 146: Zn, Co, and Ni are all divalent in this work. Is there any evidence to justify electrostatics as the most likely explanation for the observed differences in the water network?

Page 8, line 163: I think that it is backwards to say that Zn fine-tunes the structural arrangements. Zn is an inert ion. The protein scaffold is the thing which has been fine tuned by evolution to utilize the intrinsic properties of Zn to perform the relevant reaction.

Firstly, we thank the editor and the reviewers for the careful reading of our manuscript and the positive and constructive comments. Below, the reviewers' comments are addressed, with our point-by-point responses in bold.

Reviewer #1 (Remarks to the Author):

The manuscript by J. K. Kim et al. provides many new crystal structures of different metal variants of human carbonic anhydrase II with and without a substrate molecule captured in its catalytic center. Specific knowledge and techniques were employed since the catalyzed reaction of the analyzed enzyme is very fast and the substrate is gaseous. The authors analyzed metal coordination geometry, its effect on catalytic activity and water molecule network. The results will be interesting and useful to the carbonic anhydrase research field, but also could be used for better understanding of other metalloenzyme systems. I believe the information could be employed for more precise computer simulations/ docking procedures of small molecule inhibitors for metalloenzymes. The manuscript is clearly written, with adequate citation and optimal manuscript length. I would suggest adding a short discussion about a possible alternative catalytic activity that may emerge for different metallovariants of the enzyme, as was recently reported by the same group (DOI: 10.1107/S2052252520000986) for copper-CA II.

To address the reviewer's pertinent suggestion, we have added a brief description and two new references on the alternative catalytic activities of the CA metallovariants on page 4 of the revised manuscript as below (highlighted in blue).

"The zinc ion can be substituted by other physiologically relevant transition metal ions such as Co²⁺, Ni²⁺, Cu²⁺, Cd²⁺, and Mn²⁺ which results in drastic changes in the catalytic activity of CA II (~ 50% active to completely inactive)²¹. It has been also reported that the metal substitutions may induce alternative catalytic activities of CA II other than CO₂/HCO₃⁻ conversion²⁷, for instance, reduction of nitrite to nitric oxide in presence of copper.²⁸"

27 Piazzetta, P., Marino, T., Russo, N. & Salahub, D. R. The role of metal substitution in the promiscuity of natural and artificial carbonic anhydrases. *Coordination Chemistry Reviews* 345, 73-85, doi:10.1016/j.ccr.2016.12.014 (2017).

28 Andring, J. T., Kim, C. U. & McKenna, R. Structure and mechanism of copper-carbonic anhydrase II: a nitrite reductase. *IUCrJ* 7, 287-293, doi:10.1107/S2052252520000986 (2020).

Some comments: authors state (lines 18-21): “Through a comparative study on the intermediate states of the zinc-bound native human CA II and non-native metal-substituted CA IIs, we demonstrate that the characteristic metal ion coordination geometries <...> directly modulate the catalytic efficacy”. Then the lines 66-68 reference to Krishnamurthy et al. 2008 review for results on percentage of catalytic activity of different metal variants CA (would it be more precise if a primary source for these results was indicated?).

As the reviewer suggested, we have now replaced Krishnamurthy et al. (2008) with the two primary sources in the revised manuscript:

“To study the role of metal ions in CA II, we selected four divalent transition-metal ions (Zn²⁺, Co²⁺, Ni²⁺, Cu²⁺) that induce drastic changes in CA II activity (100%, ~ 50%, ~ 2%, and 0%, respectively)^{21,31,32}”

~~21 — Krishnamurthy, V. M. et al. Carbonic anhydrase as a model for biophysical and physical-organic studies of proteins and protein-ligand binding. *Chem Rev* 108, 946–1051, doi:10.1021/cr050262p (2008).~~

31 Lindskog, S. & Nyman, P. O. Metal-Binding Properties of Human Erythrocyte Carbonic Anhydrases. *Biochim Biophys Acta* 85, 462-474, doi:10.1016/0926-6569(64)90310-4 (1964).

32 Coleman, J. E. Metal ion dependent binding of sulphonamide to carbonic anhydrase. *Nature* 214, 193-194, doi:10.1038/214193a0 (1967).

In methods section lines 195-197 state “The loss of enzyme activity was verified using kinetic studies. The enzyme activity was revived by an addition of 1 mM ZnCl₂.” However, no experimental data on enzymatic activity are provided in this manuscript. Could the authors describe the method used to quantify enzymatic activity and provide raw data of the enzymatic activity in the Supplementary material.

To gladly comply with the reviewer’s suggestion, we have added a detailed description of the method in the Methods section (page 9-10), and the data of the measured enzymatic activities in Supplementary Information:

“Esterase Kinetic Assay

The CO₂/HCO₃⁻ conversion catalytic activity of CA II can be measured directly by stopped flow assays, monitoring labeled CO₂/HCO₃⁻ conversion using mass

spectroscopy, or indirectly by monitoring the innate esterase activity spectroscopically^{37,38}. In this study, the esterase activity assays were performed as a control to ensure zinc was fully chelated from recombinant CA II. The 4-nitrophenyl acetate molecule is cleavable by CA II and thus used here as a colorimetric substrate. CA II cleaves the ester bond of 4-nitrophenyl acetate generating 4-nitrophenol, which is spectroscopically absorbent at 348 nm in the UV-visible spectrum. Thus, the reaction can be monitored spectroscopically at 348nm³⁹.

In a 96 deep-well plate, aliquots of 50 μ L of 0.1 mg/mL CA II in storage buffer were added to each well. To initiate the reaction, 200 μ L of 0.8 mM 4-NPA dissolved in 3% acetone in water was added to the sample well. The well plate was then immediately inserted into the plate reader (Synergy HTX, BioTek, Winooski, WI, USA). Absorbance at 348 nm was recorded every 8 s for 10 min. The absorbance data of Apo- and Zn-CA II are plotted in Supplementary Fig. 4.

- 37 Coleman, J. E. Mechanism of Action of Carbonic Anhydrase - Substrate Sulfonamide and Anion Binding. *Journal of Biological Chemistry* 242, 5212-+ (1967).
- 38 Tu, C. & Silverman, D. Catalysis of cobalt (II)-substituted carbonic anhydrase II of the exchange of oxygen-18 between carbon dioxide and water. *Biochemistry* 24, 5881-5887 (1985).
- 39 Tashian, R. E., Douglas, D. P. & Yu, Y. S. Esterase and hydrase activity of carbonic anhydrase. I. From primate erythrocytes. *Biochem Biophys Res Commun* 14, 256-261, doi:10.1016/0006-291x(64)90445-0 (1964).

Supplementary Fig. 4. Absorbance of Apo- and Zn-CA II in esterase kinetic assay at pH 7.8. CA II esterase activity was measured spectroscopically at 348nm, indicative of substrate 4-nitrophenyl acetate hydrolysis. Compared to Zn-CA II, Apo-CA II and buffer show little to no esterase activity. The standard deviation errors (white) are presented in the data points and are ranging from 0.2 % ~ 1.1 %.

In my opinion, this manuscript is of high scientific quality and could be published in Nature communications if the requested information about the catalytic activity is provided.

Reviewer #2 (Remarks to the Author):

The paper by Kim et al. reports a detailed study by means of X-ray crystallography on the effect of metal ion substitution in the carbonic anhydrase II active site, showing that the change of coordination geometry modulates the catalytic efficiency and the network of water molecules in the enzyme active site. The paper is rather interesting, technically sound and logically arranged; however, in my opinion it does not possess sufficient novelty to be published in a so high impact factor journal such as Nature Communication.

We thank the reviewer for the kind words and would like to take this opportunity to highlight the novelty of the current manuscript. In metalloenzyme research, a long-standing problem remains as to why metalloenzymes often show dramatic changes in their activity when the native metal ion is substituted by chemically similar, but distinct non-native metal ions.

In our present work, we explicitly address this issue by investigating the roles of various metal ions in the carbonic anhydrase II (CA II) catalysis. Our results clearly show that the characteristic metal coordination geometries can directly modulate the catalytic mechanism, and that the metal ions have an unexpected long-range (< 10 Å) electrostatic influence on fine-tuning the catalytically important water network in the active site. These observations have been obtained at unprecedented structural detail, which, in our opinion, are novel to merit as significant advancement in understanding the functioning of metalloenzymes, and CA catalysis, in particular.

Reviewer #3 (Remarks to the Author):

This work seeks to systematically evaluate the contribution of the catalytic metal of human carbonic anhydrase II using X-ray crystallography. Sixteen high-resolution structures were obtained under various conditions from apo-enzyme to Zn, Co, Ni, or Cu loaded with and without CO₂ pressure at pH 7.8 or 11. This manuscript represents a substantial undertaking which provides high-quality, information-dense data sets for the metallobiochemistry field. My major concern is that the compactness of the presentation may lessen impact for readers who are not already intimately familiar with the CA II system. Also, some conclusions (detailed below) are drawn too definitely considering that this study does not include any biochemical or enzymological characterizations.

We thank the reviewer for pointing out important suggestions for the improvement of our manuscript. We agree with the reviewer's concerns and have gladly accepted the reviewer's suggestions and revised the manuscript accordingly.

First of all, for biochemical characterizations, we have included a detailed description of the characterization method in the Methods section (page 9-10), and the data of the measured enzymatic activities in Supplementary Information:

"Esterase Kinetic Assay

The CO₂/HCO₃⁻ conversion catalytic activity of CA II can be measured directly by stopped flow assays, monitoring labeled CO₂/HCO₃⁻ conversion using mass spectroscopy, or indirectly by monitoring the innate esterase activity spectroscopically^{37,38}. In this study, the esterase activity assays were performed as a control to ensure zinc was fully chelated from recombinant CA II. The 4-nitrophenyl acetate molecule is cleavable by CA II and thus used here as a colorimetric substrate. CA II cleaves the ester bond of 4-nitrophenyl acetate generating 4-nitrophenol, which is spectroscopically absorbent at 348 nm in the UV-visible spectrum. Thus, the reaction can be monitored spectroscopically at 348nm³⁹.

In a 96 deep-well plate, aliquots of 50 µL of 0.1 mg/mL CA II in storage buffer were added to each well. To initiate the reaction, 200 µL of 0.8 mM 4-NPA dissolved in 3% acetone in water was added to the sample well. The well plate was then immediately inserted into the plate reader (Synergy HTX, BioTek, Winooski, WI, USA). Absorbance at 348 nm was recorded every 8 s for 10 min. The absorbance data of Apo- and Zn-CA II are plotted in Supplementary Fig. 4.

- 37 Coleman, J. E. Mechanism of Action of Carbonic Anhydrase - Substrate Sulfonamide and Anion Binding. *Journal of Biological Chemistry* 242, 5212-5219 (1967).
- 38 Tu, C. & Silverman, D. Catalysis of cobalt (II)-substituted carbonic anhydrase II of the exchange of oxygen-18 between carbon dioxide and water. *Biochemistry* 24, 5881-5887 (1985).
- 39 Tashian, R. E., Douglas, D. P. & Yu, Y. S. Esterase and hydrase activity of carbonic anhydrase. I. From primate erythrocytes. *Biochem Biophys Res Commun* 14, 256-261, doi:10.1016/0006-291x(64)90445-0 (1964).

Supplementary Fig. 4. Absorbance of Apo- and Zn-CA II in esterase kinetic assay at pH 7.8. CA II esterase activity was measured spectroscopically at 348nm, indicative of substrate 4-nitrophenyl acetate hydrolysis. Compared to Zn-CA II, Apo-CA II and buffer show little to no esterase activity. The standard deviation errors (white) are presented in the data points and are ranging from 0.2 % ~ 1.1 %.

Major Concerns:

Figure 2 condenses too much information and too many ideas. It contains information about i) changes in coordination geometry upon metal binding/substitution (a-e), ii) geometry of CO₂ binding (f,g,h), iii) geometry of HCO₃ binding (i,m), iv) how pH affects CO₂ vs HCO₃ binding (h,m), v) effect of Zn on CO₂ binding (k), and iv) steric clashes of hypothetical CO₂ binding (n,o). While this is certainly space efficient, I think there is enough interesting data here to justify 2 or 3 separate figures with more narrow focus. If this advice is taken, consider moving panels from Supplementary figure 3 to the main text.

To present a surfeit of information condensed in Figure 2 in a more digestible manner, we have rearranged the figure into four new figures (Figures 2, 3, 4 and 5) in the revised manuscript. We also transferred Supplementary Figure 3 to the main text, incorporating it into Fig. 4 as suggested. Below are the new labeled Figures 2-5 and the corresponding captions.

From

Fig 2. Active site of CA II showing metal coordination and substrate/product binding at 0 atm and 20 atm of CO₂ pressure. The intermediate water (W_1) is colored in steel blue for clarity. The electron density ($2F_o - F_c$ blue) and the difference map ($F_o - F_c$ green) are contoured at 2.2σ and 7.0σ , respectively. All structures were obtained at pH 7.8 except for **c**) and **h**) which are obtained at pH 11.0. **a-e**) Metal coordination of apo, native and non-native metal substituted CA II without CO₂ pressurization. Zn- and Co-CA II show tetrahedral, Ni-CA II octahedral, and Cu-CA II trigonal bipyramidal coordination. **f-j**) At 20 atm of CO₂ pressure, Zn-CA II shows clear binding of CO₂ as in

apo-CA II while maintaining tetrahedral metal coordination, but Co-CA II at pH 11.0 shows superposition of CO₂ binding (~ 50% occupancy) with tetrahedral coordination and HCO₃⁻ binding (~ 50% occupancy) with octahedral coordination. Ni-CA II maintains octahedral coordination with HCO₃⁻ binding, but Cu-CA II shows disordered electron density in the CO₂/HCO₃⁻ binding site. **k**) Comparison of CO₂ binding at apo-CA II and Zn-CA II (white). **l**) Upon CO₂ binding (white) in Zn-CA II, W_{Zn} is located at the center of hypothetical tetrahedral structure made up of Zn²⁺ ion, Thr199-O2, position (1) (close to W1), and position (2) (close to the carbon atom in CO₂). In this configuration, a hybridized lone pair in W_{Zn} directly faces CO₂ molecule at a proper distance, which is required for efficient nucleophilic attack. **m**) Co-CA II at pH 7.8 shows full binding of HCO₃⁻, showing octahedral coordination. **n-o**) Compared to the W_{Zn} geometry in **l**), the nucleophilic attack geometries of W_{Ni} and W_{Cu} have steric hindrance on CO₂ molecule (from Zn-CA II, 20 atm) and are distorted away. Distance between the position (2) and C atom of CO₂ is 0.36, 1.55 and 2.93 Å for Zn-, Ni- and Cu-CA II, respectively.

Supplementary Fig. 3. Substrate and product binding in Co-CA II at pH 7.8 (a-b) and 11.0 (c-d). At pH 11.0, the active site shows dual binding of CO₂ and HCO₃⁻ when cryocooled under 20 atm CO₂ pressurization. However, at pH 7.8, the active site shows full binding of HCO₃⁻ molecule even when the Co-CA II crystal is untreated with CO₂ gas. It is likely that the captured HCO₃⁻ is converted from the CO₂ absorbed in the crystal from air. The electron density ($2F_o - F_c$, blue) and the difference map ($F_o - F_c$, green) are contoured at 2.2σ and 7.0σ , respectively. The intermediate water (W₁) is colored in steel blue for clarity.

To

Fig. 2. Metal coordination geometry in CA II without CO₂ pressurization. **a)** In apo-CA II, the metal binding site is vacant. **b-c)** Zn- and Co-CA II show tetrahedral, **d)** Ni-CA II octahedral, and **e)** Cu-CA II trigonal bipyramidal coordination geometry. The electron density ($2F_o - F_c$ blue) is contoured at 2.2σ . All structures were obtained at pH 7.8 except for **c)** which is obtained at pH 11.0. The intermediate water (W_i) in **d)** is colored in steel blue for clarity.

Fig. 3. Substrate/product binding in apo- and Zn-CA II. The intermediate water (W_1) is colored in steel blue for clarity. The electron density ($2F_o - F_c$, blue) is contoured at 2.2σ . **a-b)** At 20 atm of CO_2 pressure, apo-CA II shows clear binding of CO_2 without the need of Zn^{2+} ion. **c)** Zn-CA II shows similar binding of CO_2 as in apo-CA II while maintaining tetrahedral metal coordination. **d)** Upon CO_2 binding (white) in Zn-CA II, W_{Zn} is located at the center of the hypothetical tetrahedral arrangement made up of Zn^{2+} ion, Thr199-O2, position (1) (close to W_1), and position (2) (close to the carbon atom in CO_2). In this configuration, a hybridized lone pair in W_{Zn} directly faces CO_2 molecule at a distance, appropriate for efficient nucleophilic attack. Distance between the position (2) and C atom of CO_2 is merely 0.36 Å.

Fig. 4. Substrate/product binding in Co-CA II. The intermediate water (W_1) is colored in steel blue for clarity. The electron density ($2F_o-F_c$, blue) and the difference map (F_o-F_c , green) are contoured at 2.2σ and 7.0σ , respectively. **a-b**) At 20 atm of CO_2 pressure, Co-CA II at pH 11.0 shows superposition of CO_2 binding ($\sim 50\%$ occupancy) with tetrahedral coordination and HCO_3^- binding ($\sim 50\%$ occupancy) with octahedral coordination. **c-d**) Co-CA II at pH 7.8 shows complete binding of HCO_3^- , showing octahedral coordination even in absence of added CO_2 . It is likely that the captured HCO_3^- is converted from the CO_2 absorbed in the crystal from ambient air.

Fig. 5. Substrate/product binding in Ni- and Cu-CA II. The intermediate water (W_1) is colored in steel blue for clarity. The electron density ($2F_o - F_c$, blue) is contoured at 2.2σ . **a**) At 20 atm of CO_2 pressure, Ni-CA II maintains octahedral coordination with HCO_3^- binding. **b**) Compared to the W_{Zn} geometry in Zn-CA II (Fig. 3d), the nucleophilic attack geometry of W_{Ni}' has steric hindrance on CO_2 molecule (adapted from Zn-CA II, 20 atm) and is distorted away. Distance between the position (2) and C atom of CO_2 is 1.55 Å. **c**) Cu-CA II shows only disordered electron density in the CO_2/HCO_3^- binding site. **d**) The nucleophilic attack geometry of W_{Cu} has steric hindrance on CO_2 molecule (adapted from Zn-CA II, 20 atm) and is significantly distorted away. Distance between the position (2) and C atom of CO_2 is 2.93 Å.

Page 5, lines 89-95: The authors note that carbonate binds Co-CA II in a bidentate fashion, leading to a higher binding affinity which in turn reduces overall activity. While bidentate rather than monodentate binding may correlate with a higher binding affinity, it is not sufficient to draw such a conclusion, especially regarding the solution state behavior of an enzyme.

In the native CA II, the rate limiting step of the overall catalytic reaction is the proton transfer process. If the rate limiting step in the Co-CA II remains same as in the native CA II, we agree that the stronger bidentate binding does not necessarily reduce the overall activity. Accordingly, we removed the conclusive statement in the revised manuscript as:

"In the transformed octahedral geometry, the HCO_3^- molecule is bound in a bidentate mode to the Co^{2+} ion along with an additional water molecule. Compared to the monodentate binding mode in Zn-CA II, the negative charge on the bidentate HCO_3^- can be distributed among the two oxygen atoms bound to Co^{2+} ion, allowing stronger product binding to the metal ion (Supplementary Fig.1a-f). ~~The higher affinity for HCO_3^- in Co-CA II seems related to its lower catalytic activity, as HCO_3^- is more difficult to displace to repeat the catalytic cycle.~~"

Supplementary Fig. 1c&f: Unless proton positions have been experimentally determined (Neutron diffraction or spectroscopically) or supported by computational efforts, it is best to leave their positions unassigned.

To ensure rapid CA catalytic activity, it is essential that the negatively charged bicarbonate is released readily from the positively charged Zn^{2+} ion. Researchers on CA have studied possible binding configurations of bicarbonate to Zn^{2+} ion, and suggested that the specific hydrogen bonding between the monodentate bicarbonate and the OH group of Thr199 (as shown in Supplementary Fig 1c) is essential for the facile release of bicarbonate (the specific configuration shifts the negative charge within the bicarbonate away from the metal ion). This suggestion was supported by the mutational studies on Thr199, in which the catalytic activity of a single Thr199 mutant decreased significantly to just a few fractions of its native form. Thus, the bicarbonate binding configuration in Supplementary Fig 1c has been supported experimentally and is widely accepted in the CA research community. In addition, we believe that the bicarbonate binding configuration in Supplementary Fig 1f could serve as a reference

configuration for the study on the interactions between the bicarbonate and the Co^{2+} and Ni^{2+} ions. Therefore, we wish to retain the Supplementary Figs 1c&f as they stand. However, to help readers grasp this point more clearly, we have added 3 relevant references in the revised manuscript as below.

“Fig. 1.

.....

c) The CO_2 hydration reaction mechanism of Zn-CA II. First, CO_2 binds to the active site, leading to a nucleophilic attack by the zinc-bound hydroxyl ion onto CO_2 . HCO_3^- thus formed is subsequently displaced by the water molecule inflowing through EC. The HCO_3^- molecule likely binds to Zn^{2+} ion in a monodentate mode and its OH group is held at the Zn^{2+} ion due to the hydrogen bonding with Thr199^{52,53}. This product binding configuration leads to a weak interaction between the product and Zn^{2+} ion, thereby facilitating fast product dissociation⁵⁴. Finally, proton transfer occurs via the network ($\text{W}_{\text{Zn}} \rightarrow \text{W1} \rightarrow \text{W2} \rightarrow \text{His64}$) provided by the protein scaffold.”

52 Krebs, J. F., Ippolito, J. A., Christianson, D. W. & Fierke, C. A. Structural and functional importance of a conserved hydrogen bond network in human carbonic anhydrase II. *J Biol Chem* 268, 27458-27466 (1993).

53 Xue, Y., Liljas, A., Jonsson, B. H. & Lindskog, S. Structural analysis of the zinc hydroxide-Thr-199-Glu-106 hydrogen-bond network in human carbonic anhydrase II. *Proteins* 17, 93-106, doi:10.1002/prot.340170112 (1993).

54 Liljas, A. Carbonic anhydrase under pressure. *IUCrJ* 5, 4-5, doi:10.1107/S2052252517018012 (2018).

Page 6, line 117: Avoid using “This” as the subject of a sentence. Also, adding a qualifier to this sentence seems appropriate.

As the reviewer suggested, we have changed the corresponding sentence in the revised manuscript as below.

“~~This explains~~The inefficient substrate binding and the unfavorable distorted geometry explain the complete enzymatic inactivity of Cu-CA II. “

The second and third column of Figure 3 appear to show different angles of the same data set. Such presentation is usually accompanied by a curved arrow labelling the degree of rotation. In this case, following convention would also reduce the number of panel labels needed by 5. Also, the degree to which the conformation of His64 is altered does not seem significant enough to warrant inclusion in the figure. These values may be better presented in a table in the SI.

Based on the reviewer's comments, we have modified Fig. 3 and relabeled it as Fig. 6 in the revised manuscript. We have also transferred the His64 conformation values to Supplementary Table 1.

From

Fig. 3. Active site in CA II showing proton transfer pathway and EC water network ($W_{EC1} \sim W_{EC5}$).

To

Fig. 6. Active site in CA II showing proton transfer pathway and EC water network (W_{EC1} ~ W_{EC5}).

Supplementary Table 1. Data collection and refinement statistics for the CA II structures.

	apo-CA II 0atm 6LUU	apo-CA II 20atm 6LUV	Zn-CA II 0atm 6LUW	Zn-CA II 20atm 6LUX	Zn-CA II 0atm pH11.0 6LUY	Zn-CA II 20atm pH11.0 6LUZ	Co-CA II 0atm 6LV1	Co-CA II 20atm 6LV2
Data collection								
Space group	$P2_1$	$P2_1$	$P2_1$	$P2_1$	$P2_1$	$P2_1$	$P2_1$	$P2_1$
Cell dimensions								
a, b, c (Å)	42.13, 41.30, 72.21	42.26, 41.38, 72.00	42.21, 41.28, 72.15	42.37, 41.44, 72.13	42.28, 41.26, 72.07	42.39, 41.47, 72.15	42.31, 41.22, 72.05	42.32, 41.32, 72.22
β (°)	104.27	104.18	104.18	104.05	104.19	104.11	104.16	104.03
Resolution (Å)	30-1.20 (1.22-1.20)	30-1.20 (1.22-1.20)	30-1.20 (1.22-1.20)	30-1.20 (1.22-1.20)	30-1.20 (1.22-1.20)	30-1.20 (1.22-1.20)	30-1.20 (1.22-1.20)	30-1.20 (1.22-1.20)
R_{sym} (%)	5.8 (36.8)	7.1 (48.1)	6.5 (37.6)	6.8 (66.9)	8.5 (17.7)	5.6 (20.3)	9.2 (60.7)	8.2 (65.6)
$I/\sigma(I)$	28.8 (6.8)	21.2 (5.1)	29.7 (6.3)	29.8 (3.6)	25.2 (12.7)	29.9 (11.8)	24.3 (3.2)	20.1 (3.8)
Completeness (%)	95.5 (92.6)	94.3 (91.2)	98.8 (97.5)	96.2 (92.9)	98.0 (96.0)	94.6 (91.9)	96.0 (93.3)	98.4 (96.4)
Redundancy	7.5 (7.3)	7.5 (7.4)	7.3 (7.1)	7.4 (7.3)	7.4 (7.4)	7.6 (7.5)	7.6 (7.6)	7.4 (7.2)
Refinement								
Resolution (Å)	1.20	1.20	1.20	1.20	1.20	1.20	1.20	1.20
No. reflections	72,011	71,305	74,561	73,183	73,898	72,112	72,478	74,640
$R_{\text{work}}/R_{\text{free}}$ (%)	11.0 / 13.8	11.7 / 14.8	11.7 / 14.1	11.3 / 14.2	11.1 / 13.1	10.3 / 12.7	12.0 / 15.0	11.4 / 13.9
No. atoms								
Protein	4,285	4,256	4,253	4,245	4,253	4,245	4,259	4,240
Ligand/ion	1 glycerol	2 CO ₂ ,	1 glycerol	2 CO ₂ ,	1 glycerol	2 CO ₂ ,	1 HCO ₃ ⁻ ,	1 HCO ₃ ⁻ ,
Water		1 glycerol		1 glycerol		1 glycerol	1 glycerol	1 CO ₂ ,
B -factors								1 glycerol
Protein (main / side chain)	371	373	263	367	264	365	283	333
Ligand/ion	9.78 / 12.75	11.48 / 14.38	10.09 / 13.22	10.80 / 13.62	7.90 / 10.99	8.87 / 11.51	10.84 / 14.07	10.31 / 13.31
Water	20.76 (glycerol)	13.90 (first CO ₂),	18.32 (glycerol)	10.63 (first CO ₂),	14.79 (glycerol)	8.80 (first CO ₂),	11.23 (HCO ₃ ⁻),	9.25 (HCO ₃ ⁻),
Ligand/ion		26.78 (second CO ₂),		23.50 (second CO ₂),		21.13 (second CO ₂),	24.46 (glycerol)	23.17 (second CO ₂),
Water		18.39 (glycerol)		17.17 (glycerol)		14.78 (glycerol)		21.27 (glycerol)
R.m.s. deviations	30.25	30.67	26.12	30.58	24.33	27.75	27.41	29.71
Bond lengths (Å)	0.031	0.030	0.029	0.028	0.029	0.028	0.031	0.028
Bond angles (°)	2.515	2.435	2.437	2.423	2.506	2.368	2.644	2.284
Partial occupancy (%)								
His64 (out/in) conformation (%)	58 / 42	44 / 56	40 / 60	69 / 31	32 / 68	63 / 37	49 / 51	68 / 32
	Co-CA II 0atm pH11.0 6LV3	Co-CA II 20atm pH11.0 6LV4	Ni-CA II 0atm 6LV5	Ni-CA II 20atm 6LV6	Ni-CA II 0atm pH11.0 6LV7	Ni-CA II 20atm pH11.0 6LV8	Cu-CA II 0atm 6LV9	Cu-CA II 20atm 6LVA
Data collection								
Space group	$P2_1$	$P2_1$	$P2_1$	$P2_1$	$P2_1$	$P2_1$	$P2_1$	$P2_1$

Cell dimensions								
a, b, c (Å)	42.33, 41.26, 72.08	42.36, 41.46, 72.31	42.40, 41.29, 71.92	42.40, 41.37, 72.13	42.52, 41.23, 71.87	42.42, 41.40, 72.19	42.33, 41.23, 72.09	42.36, 41.40, 72.31
β (°)	104.13	104.02	104.03	104.01	104.06	104.04	104.17	103.97
Resolution (Å)	30-1.20 (1.22-1.20)	30-1.20 (1.22-1.20)	30-1.20 (1.22-1.20)	30-1.20 (1.22-1.20)	30-1.20 (1.22-1.20)	30-1.20 (1.22-1.20)	30-1.20 (1.22-1.20)	30-1.20 (1.22-1.20)
R_{sym} (%)	8.5 (46.4)	5.8 (42.4)	8.6 (33.1)	7.2 (37.7)	8.5 (29.7)	4.7 (22.7)	9.9 (42.8)	7.8 (44.8)
$I/\sigma(I)$	23.0 (4.3)	31.8 (5.6)	20.7 (7.6)	23.1 (7.2)	23.1 (8.4)	40.6 (10.1)	19.5 (6.0)	25.3 (5.5)
Completeness (%)	95.0 (91.7)	94.6 (91.2)	96.6 (93.8)	95.4 (92.1)	97.5 (94.7)	96.1 (93.0)	95.1 (92.3)	96.9 (94.3)
Redundancy	7.6 (7.7)	7.6 (7.6)	7.5 (7.4)	7.6 (7.5)	7.4 (7.3)	7.5 (7.3)	7.7 (7.6)	7.5 (7.5)
Refinement								
Resolution (Å)	1.20	1.20	1.20	1.20	1.20	1.20	1.20	1.20
No. reflections	71,822	72,158	73,007	72,507	73,791	73,217	71,843	73,841
$R_{\text{work}}/R_{\text{free}}$ (%)	12.3 / 14.9	11.3 / 13.9	11.6 / 14.1	11.2 / 13.8	11.7 / 13.9	11.2 / 13.6	11.7 / 14.4	11.3 / 14.2
No. atoms								
Protein	4,252	4,240	4,259	4,240	4,259	4,240	4,259	4,240
Ligand/ion	1 glycerol	1 HCO ₃ ⁻ ,	1 glycerol	1 HCO ₃ ⁻ ,	1 glycerol	1 HCO ₃ ⁻ ,	1 glycerol	1 CO ₂ ,
Water	265	1 glycerol 333	279	1 glycerol 347	279	1 glycerol 346	286	360
B-factors								
Protein (main / side chain)	10.84 / 14.06	10.39 / 13.16	8.73 / 11.74	7.93 / 10.50	9.15 / 12.17	8.38 / 10.83	9.71 / 12.55	8.23 / 10.89
Ligand/ion	21.51 (glycerol)	15.78 (HCO ₃ ⁻), 14.19 (first CO ₂),	18.86 (glycerol)	9.15 (HCO ₃ ⁻), 21.92 (second CO ₂),	19.13 (glycerol)	7.40 (HCO ₃ ⁻), 21.95 (second CO ₂),	17.19 (glycerol)	19.67 (second CO ₂), 14.81 (glycerol)
Water	26.98	21.78 (second CO ₂), 19.39 (glycerol)	24.56	14.87 (glycerol)	26.51	13.93 (glycerol)	26.62	29.36
R.m.s. deviations								
Bond lengths (Å)	0.031	0.028	0.032	0.029	0.030	0.028	0.031	0.029
Bond angles (°)	2.484	2.358	2.600	2.382	2.552	0.352	2.550	2.460
Partial occupancy (%)								
His64 (out/in conformation (%))	49 / 51	69 / 31	43 / 57	60 / 40	46 / 54	52 / 48	39 / 61	66 / 34

*Values in parentheses are for the highest-resolution shell.

Minor Concerns:

The abstract mentions “unprecedented roles of metal ions in a model metalloenzyme system,” however I did not find anything unprecedented about the roles of the metal ions in this work.

We removed the word “unprecedented” and inserted the word “catalytic” in the revised abstract:

“Here we report ~~the unprecedented~~ on the catalytic roles of metal ions in a model metalloenzyme system, carbonic anhydrase II (CA II).”

Fig. 1c is not referenced until the discussion. It should be discussed before Fig 2 or moved to a later figure.

We thank the reviewer for pointing this out. As suggested by the reviewer, Fig. 1c is now cited before Fig. 2 in the introduction of the revised manuscript (page 3) by inserting the blue parentheses as below:

“It possesses a well-defined active site containing a single metal-binding site (Fig. 1a-b), and the kinetic rates and fine details of the enzymatic mechanism have been studied extensively¹⁷⁻²¹ ~~(Fig. 1)~~(Fig. 1c).”

Page 5, lines 78 & 88: It should be mentioned that the Co-CA II structure (Fig. 2c) was obtained under a different condition (pH 11.0) than all other structures (pH 7.8).

As suggested by the reviewer, the pH value of the Co-CA II structure was included in the revised manuscript (new insertions marked in blue):

“

In Zn-CA II and Co-CA II (pH 11.0), the metal ions display tetrahedral coordination with three histidine residues (His94, His96, and His119) and a water molecule (Fig. 2b-c).

...

In Co-CA II (pH 11.0) cryocooled at 20 atm CO₂ pressure, dual binding of CO₂ and HCO₃⁻ is observed (Fig. 4a).”

Page 6, lines 108,9: Deprotonation of the metal-bound water is proposed as cause of carbonate dissociation in the case of Co. Can the authors provide any insight as to why that same reasoning cannot be applied to Ni?

Our crystallographic study shows that the bicarbonate binding affinity in Co-CA II decreases as pH increases, but that in Ni-CA II is almost inert to pH variations. At high pH value such as 11.0, the metal-bound water molecule can be easily deprotonated forming a negatively charged hydroxide ion. Considering the charge-charge repulsion, this negatively charged hydroxide ion can destabilize and dissociate the metal-bound negatively charged bicarbonate. We believe that this charge-charge repulsion is operative in Co-CA II, and as consequently, upon bicarbonate dissociation, the transient octahedral coordination geometry reverts to the more stable tetrahedral coordination geometry (Fig. 7a). However, in Ni-CA II, it appears that such charge-charge repulsion is insufficient to dissociate the bicarbonate molecule as the octahedral coordination is maintained by Ni²⁺ ion throughout the entire catalytic cycle and the bicarbonate seems stabilized within such configuration (Fig. 7b). We believe that a more conclusive interpretation requires elaborate computational efforts at the quantum chemistry level. However, to guide readers, we have added a brief description on the charge-charge repulsion in the revised manuscript (page 5-6) as below.

“

Unlike Zn-CA II (Supplementary Fig. 2), the Co-CA II intermediates obtained at different pH values (7.8 and 11.0) reveal that the HCO₃⁻ molecule is firmly bound to Co²⁺ ion with full occupancy at lower pH (Fig. 4c-d), but this binding affinity weakens as pH increases (Fig. 4a). ~~The result suggests that, during the catalytic cycle, the bound HCO₃⁻ molecule may be dissociated from the Co²⁺ ion following deprotonation of the Co²⁺-bound water, forming negatively charged hydroxide ion.~~ The result suggests that, during the catalytic cycle, deprotonation of the Co²⁺-bound water may lead to dissociation of the HCO₃⁻ molecule from the Co²⁺ ion, due to the charge-charge repulsion between the formed hydroxide ion and the HCO₃⁻ molecule. Following the HCO₃⁻ dissociation, the tetrahedral coordination is restored (Fig. 2c).

...

Unlike Co-CA II (Fig. 4a & 4c-d), the Ni-CA II intermediates obtained at different pH values (7.8 and 11.0) indicate that the HCO₃⁻ binding affinity is almost unresponsive to

pH variation (Supplementary Fig. 3). ~~The result suggests that the bound HCO_3^- is directly displaced by two incoming water molecules in Ni-CA II, independent of the deprotonation of the Ni^{2+} -bound water.~~ The result suggests that the deprotonation of the Ni^{2+} -bound water is insufficient to facilitate HCO_3^- dissociation in the stable octahedral coordination, and that the bound HCO_3^- is directly displaced by two incoming water molecules in Ni-CA II."

Page 7, line 131: If it is a fact that Zn produces a long-range electrostatic field, then an appropriate citation should be provided.

To the best of our knowledge, the long-range electrostatic field effect on water network in a metalloenzyme has not been reported before. It has been suggested based on our current experimental results and has no precedent. In light of this fact, we modified the corresponding sentence in the revised manuscript as below.

"These observations ~~hint at the fact suggest~~ that the Zn^{2+} ion produces a long-range (~10 Å) electrostatic field in which water structure and dynamics in the active site are fine-tuned to facilitate the proton transfer and the water/substrate/product exchange."

Page 7, line 146: Zn, Co, and Ni are all divalent in this work. Is there any evidence to justify electrostatics as the most likely explanation for the observed differences in the water network?

Considering that metal ions are charged entities and water molecules are highly dipolar, it seems intuitive that the interactions between a metal ion and water molecules in CA II ought to be governed by electrostatic interactions. When a metal ion is incorporated at the active site of CA II, it closely interacts with the 3 histidine residues (His94, His96, His119) and some bound water molecules. These interactions can cause distortions in the electron quantum states and the net charge distribution of the metal ion. And the degree of distortions seems non-linearly amplified by the slight differences in the divalent metal ions. Revealing the precise mechanism for such distortion is beyond the scope of our manuscript. However, we believe that our present structural insights would provide a starting point for computational studies that can address the intricacies of interaction operating at the quantum mechanical level.

Page 8, line 163: I think that it is backwards to say that Zn fine-tunes the structural arrangements. Zn is an inert ion. The protein scaffold is the thing which has been fine-tuned by evolution to utilize the intrinsic properties of Zn to perform the relevant reaction.

We thank the reviewer for raising this issue. Our original effort was aimed at addressing the water network, rather than the structural features of the protein scaffold. To clarify this point, we have modified the corresponding sentence in the revised manuscript as below.

“The most efficient native Zn-CA II preserves a tetrahedral coordination and fine-tunes ~~the structural arrangements provided by the water network embedded within the~~ protein scaffold (Fig. 1c).”

REVIEWERS' COMMENTS:

Reviewer #1 (Remarks to the Author):

This is the second review of the manuscript by J. K. Kim et al. "Elucidating the role of metal ions in carbonic anhydrase catalysis". In the reviewed version of the manuscript, the authors provided the requested missing information about the catalytic activity of the analyzed enzymes, added more precise citations of previous data, and made more other improvements.

In my opinion, this manuscript is of high scientific quality and should be published in Nature Communications.

Reviewer #2 (Remarks to the Author):

The paper is significantly improved with respect to the original submission

Reviewer #3 (Remarks to the Author):

The authors have done a superlative job responding to all reviewer queries, and the revised manuscript is acceptable for publication in Nature Communications.